
# Annual river dataset in China: a new product with a 10 m spatial resolution from 2016 to 2023

Kaifeng Peng[1,2], Beibei Si[1], Weiguo Jiang[2], Meihong Ma[1], Xuejun Wang[1,3,*]

[1]Faculty of Geography, Tianjin Normal University, Tianjin, 300387, China

[2]State Key Laboratory of Remote Sensing Science, Faculty of Geographical Science, Beijing Normal University,Beijing, 100875, China

[3] Academy of Eco-civilization Development for Jing-Jin-Ji Megalopolis, Tianjin Normal University, Tianjin, 300387, China

*Correspondence to*: Xuejun Wang (xuejunw@tjnu.edu.cn)

**Abstract**: Rivers play important roles in ecological biodiversity, shipping trade and the carbon cycle. Owing to human
disturbances and extreme climates in recent decades, river extents have altered frequently and dramatically. The development of sequential and fine-scale river extent datasets, which could offer strong data support for river protection, management and sustainable use, is urgently needed. A literature review revealed that annual river extent datasets with fine spatial resolutions are generally unavailable for China. To address this issue, the first Sentinel-derived annual China river extent dataset (CRED) from 2016 to 2023 was produced in our study. We first produced annual water maps by combining the dynamic world (DW),
ESRI global land cover (EGLC) data and the multiple index water detection rule (MIWDR). For the DW and MIWDR water time series, the mode algorithm, which calculates the most common values, was used to generate yearly water maps. Then, an object-based hierarchical decision tree based on geometric features and auxiliary datasets was developed to extract rivers from the water data. The results indicated that the overall accuracies (OAs) of the CRED were greater than 96.0% % from 2016 to 2023. The user accuracies (UAs), producer accuracies (PAs) and F1 scores of the rivers exceeded 95.3%, 91.3% and 93.7%,
respectively. A further data intercomparison indicated that our CRED shared similar patterns with the wetland map of East Asia (EA_Wetlands), China land use/cover change (CNLUCC) and China water covers (CWaC) datasets, with correlation coefficients (R) greater than 0.75. Moreover, our CRED outperformed the three datasets in terms of small river mapping and misclassification reduction. The area statistics indicated that the river area in China was 44,948.78 km$^2$ in 2023, which was mostly distributed in coastal provinces of China. From 2016 to 2023, the river areas were characterized by an initial increase,
followed by a decrease and then a slight increase. Spatially, the decreased rivers were located mainly in Southeast China, whereas the increased rivers were distributed mainly in Central China and Northeast China. In general, the CRED explicitly delineated river extents and dynamics in China, which could provide a good foundation for improving river ecology and management. The CRED dataset is publicly available at https://doi.org/10.5281/zenodo.13841910 (Peng et al., 2024a).





## 1 Introduction

30       River ecosystems are fluid water bodies with linear geometric characteristics. Known as the blood vessels of the Earth, rivers support large amounts of fresh water in natural ecosystems and human societies (Wei et al., 2020) and exert important effects on hydrological cycles and ecological evolution (Li et al., 2020). Moreover, rivers also affect the carbon trade-off in terrestrial ecosystems. As reported, global rivers emit approximately 1.8 Pg of carbon (i.e., carbon dioxide) into the atmosphere annually (Raymond et al., 2013), and the rates of these processes are importantly related to river area changes (Caissie, 2006;

Hotchkiss et al., 2015). Thus, investigating river extent changes over time, which could strongly support human well-being, dual carbon targets and hydrological modes, is necessary.

      Owing to its large imaging range and low acquisition cost, satellite technology has become an important tool for extracting rivers (Valman et al., 2024). To date, a series of river datasets have been produced at the global or national scale, which can be roughly divided into two categories: river line datasets and river polygon datasets. The river line datasets delineate river channel

lines with attributes of width, discharge and others, such as the GRRATS (Global River Radar Altimetry Time Series) (Coss et al., 2020), HydroRIVERS (Lehner and Grill, 2013), and MCRW (Multitemporal China River Width) (Yang et al., 2020a). These datasets were produced on the basis of digital elevation model (DEM) data or satellite images, which effectively display the spatial distribution of the river network. However, they cannot delineate river spatial extents and temporal area changes (Fei et al., 2022). In contrast, river polygon datasets record river complete extents as polygons. In these datasets, rivers together

with other wetland types are generally mapped on the basis of satellite images or data compilation, such as the GLWD (Global Lakes and Wetlands Database) (Lehner and Doll, 2004), GRWL (Global River Widths from Landsat) (Allen and Pavelsky, 2018), and CAS_Wetlands (Mao et al., 2020). These datasets effectively display river extents and their spatial distributions. However, a literature review revealed that multitemporal river polygon datasets at the global or national scale are still scarce, as are those for China.

50       Compared with limited river datasets, water-related products are largely produced at global and national scales (Xu et al., 2020). These datasets provide good potential for river classification because rivers are contained within water areas. Generally, water-related datasets are roughly divided into two categories: surface water datasets and land use/cover change (LUCC) datasets (Xu et al., 2022). The former directly map water extents on the basis of time series of satellite images, such as the GSW (Global Surface Water) (Pekel et al., 2016) and GSWD (Global Surface Water Dynamics) (Pickens et al., 2020) datasets.

LUCC datasets, such as NLCD (Wickham et al., 2023), GLC_FCS30D (Zhang et al., 2024), and dynamic world (DW) (Brown et al., 2022) datasets, mapped waters along with other land use types. These above datasets have good potential for large-scale river extraction.

      China has considerable river resources with world-famous rivers, such as the Yangtze River, Yellow River and Pearl River. According to China's First National Census for water, there are approximately 22,200 rivers with basin areas greater

than 100 km² (Ministry of Water Resources, 2013). However, China is one of the most water-deficient countries because of its

low per capita water resources (Bai and Zhao, 2023). Moreover, human disturbance and climate change have exacerbated river

losses in recent years (Speed et al., 2016). To balance river protection and economic development, China's government has

implemented a series of projects, such as the Yangtze River Economic Belt (She et al., 2019) and the high-quality development

of the Yellow River (Jiang et al., 2021). Considering the ecological and economic importance of rivers, it is necessary to

produce multitemporal river datasets for China.

        The objective of this study was to produce annual China river extent datasets (CRED), which, to the best of our knowledge,

are the first river datasets at a 10-m spatial resolution from 2016 to 2023 for China. To achieve this goal, we first collected two

LUCC datasets and Sentinel-2 time series to generate a yearly water map. Second, annual river extents were mapped via

geometric features and knowledge rules. The geometric features were calculated via chessboard segmentation, and the

knowledge rules were developed via sample analysis of rivers, lakes and reservoirs. Finally, test samples, created via visual

interpretation, were used to assess the river mapping accuracy. Moreover, comparisons with existing river-related datasets were

implemented to confirm the reliability of our river dataset.

**2 Study area and data**

**2.1 Study area**

In this study, China was chosen as the study area, with longitudes of 73°30'4" E and 135°5'19" E and latitudes of 6°19'24"

N and 53°33'39" N. China covers five major climate zones (e.g., tropical monsoon, subtropical monsoon, temperate monsoon,

temperate continental, and plateau mountain zones), with a temperature difference of ~30 °C and a precipitation difference of

~2,500 mm (Liu et al., 2014). In China, the distribution of rivers exhibits regional imbalance, with dense networks in the

southern and eastern regions and sparse networks in the northern and western regions (Bai and Zhao, 2023). To analyse river

characteristics in detail, we divided China into nine river basins (Fig. 1), namely, the Continental Basin (CB), Southeast Basin

(SEB), Huaihe River Basin (HuRb), Haihe River Basin (HaRB), Yellow River Basin (YeRB), Songhua and Liaohe River Basin

(SLRB), Southwest Basin (SWB), Yangtze River Basin (YaRB), and Pearl River Basin (PRB).

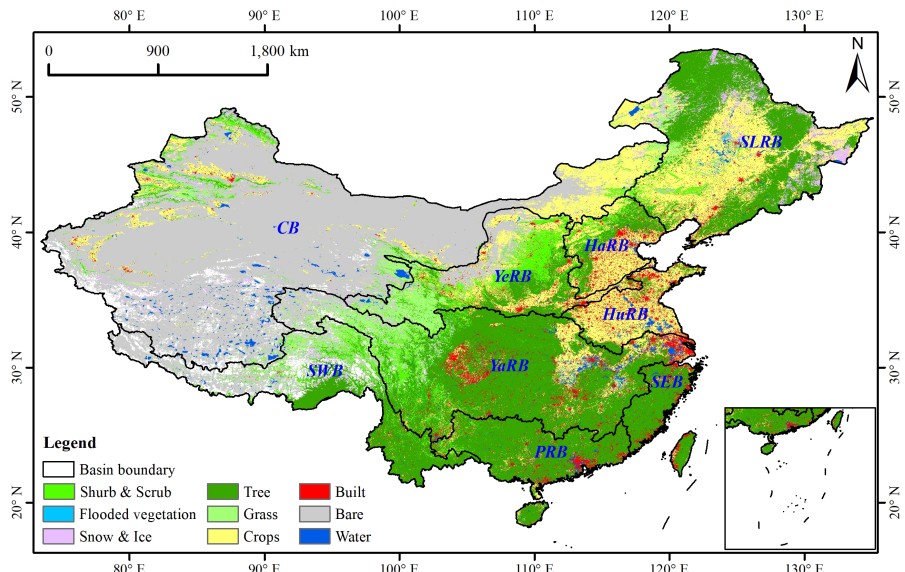

**Fig. 1. Land use patterns and river basins of China. The coloured map is a mode-composited image of DW time series images from 2023.**

### 2.2 Data collection

The dynamic world (DW), a near-real-time (NRT) land use/land cover (LULC) dataset, was selected as the primary dataset for producing water maps (Brown et al., 2022). It was produced on the basis of a Sentinel-2 time series, with a revisit interval of 2–5 days and a spatial resolution of 10 m. In the DW dataset, nine land use types were delineated, and the water areas had high mapping accuracy. Statistical analysis revealed that many regions in China had no or few good observations (Fig. S1). This occurred because the DW images were produced from Sentinel-2 images with less than 35% cloud coverage.

For areas where DW observations were missing, we used the ESRI global land cover (EGLC) dataset to supplement the data (Karra et al., 2021). The EGLC dataset was produced on the basis of Sentinel-2 images via deep learning algorithms. It recorded annual global land use types from 2017 to 2023 with high accuracy for water bodies. In addition, we also collected Sentinel-2 images to produce water maps in areas missing data from the DW dataset in 2016 because of the lack of 10-m LUCC or water datasets. All Sentinel-2 images from 2015 to 2016 were used for water classification.

To facilitate river extraction and data intercomparison, some auxiliary datasets, which are listed in Table 1, were used. The China land use/cover change (CNLUCC) dataset is a LUCC dataset that maps 23 land use types at a 30 m spatial resolution (Liu et al., 2018b). It was produced by visual interpretation, with an overall accuracy exceeding 90%. The CNLUCC of 2020 was used to produce training samples of rivers, lakes and reservoirs and make comparisons with our river maps. The Wetland map of East Asia (EA_wetlands) is a detailed wetland dataset (including rivers) of East Asia from 2021, with an overall accuracy of 88% Wang et al. (2023a). The China water cover map (CWaC) is a Chinese water cover dataset (including rivers) from 2020 with a 10 m spatial resolution and an overall accuracy of 86% (Li and Niu, 2022). EA_Wetlands and CWaC were





also used to compare the data to validate the reliability of our river maps. In addition, the dam points, including the reservoir

point of interest (POI), OSM_DEM, and GOODD (Mulligan et al., 2020), were used to distinguish reservoirs from natural

water bodies.

**Table 1. Study data and information in our study**

| Type | Name | Spatial Resolution | Time period | Data description | Data source |
|---|---|---|---|---|---|
| Land use | DW | 10 m | 2015–2023 | Water extraction | Brown et al. (2022) |
| | EGLC | 10 m | 2017–2023 | | Karra et al. (2021). |
| | CNLUCC | 30 m | 2020 | Data comparison Sample produce | Liu et al. (2018b) |
| Satellite image | Sentinel-2 | 10 m | 2015–2016 | Water extraction | Google Earth Engine (GEE) |
| Wetland dataset | EA_Wetlands | 10 m | 2021 | data comparison | Wang et al. (2023a) |
| | CWaC | 10 m | 2020 | | Li and Niu (2022) |
| Auxiliary dataset | POI | --- | 2023 | Reservoir extraction | https://lbs.amap.com |
| | GOODD | --- | --- | | Mulligan et al. (2020) |
| | OSM_Dam | --- | --- | | |

**3 Methodology**

This study aimed to produce a sequential and fine-resolution river extent dataset for China, and the processing chain

included yearly water map generation, river classification, and accuracy validation (Fig. 2). First, the mode algorithm was used

to composite yearly water map-based DW datasets. For areas with missing DW data, the EGLC and Sentinel-2 images were

chosen as supplementary datasets, which were utilized to create annual water maps. Second, an object-based hierarchical

decision tree algorithm was developed on the basis of geometric features and auxiliary datasets that can extract rivers well

from water polygons. Finally, test samples were manually produced to assess the classification accuracy. Additionally, river-

related datasets were also used to compare data to test the reliability of our river maps.

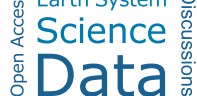

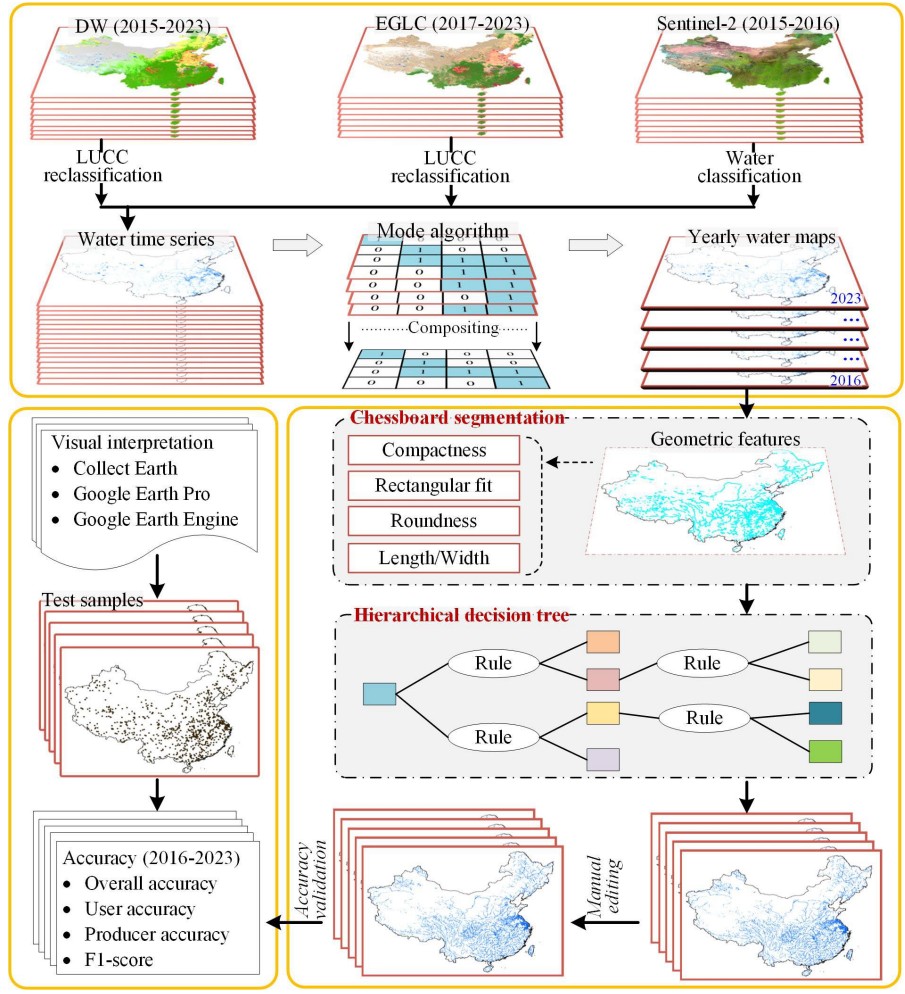

**Fig. 2. Workflow of annual river extraction**

### 3.1 Annual water generation

Rivers are distributed within the extent of surface water. Thus, water extraction can offer a good data foundation for river

mapping. To facilitate computation, we divided China into 52 regions using 5°×5° tiles. Through the statistics of valid DW

observations from 2016 to 2023, we found that many areas, which are mostly distributed in Southwest China, have a small

number or no valid land use pixels (Fig. S1). To address this issue, three scenarios were designed for water extraction (Fig. 3).

Specifically, if all the valid DW pixels in a tile exceeded three in a year, the DW dataset was selected to produce annual water

maps. Otherwise, the EGLC dataset was used to create annual maps from 2017 2023, or the Sentinel-2 images from 2015 to

2016 were used to generate water maps for 2016. In addition, the DW dataset for tiles 49, 51 and 52 had poor water accuracies.

For the three tiles, in our study, annual water maps from 2017-2023 were created via the EGLC dataset, whereas those for

2016 were produced via Sentinel-2 images.

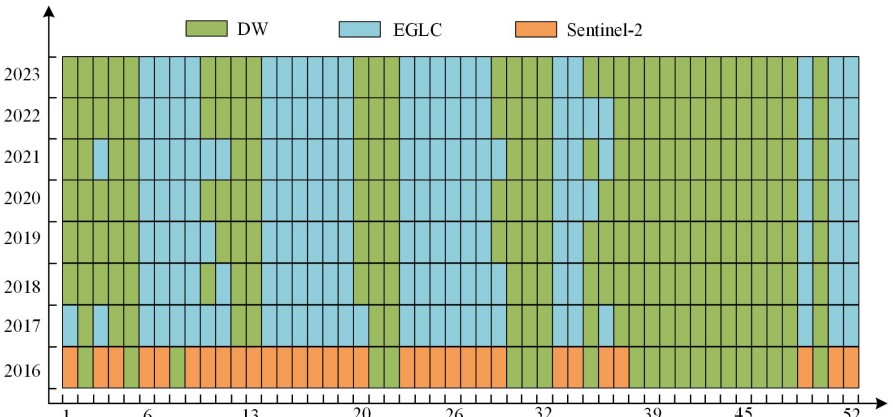

**Fig. 3. Three scenarios for water extraction from different datasets**

**3.1.1 Water extraction based on the DW dataset**

The DW dataset provides mapped water areas along with eight additional land cover types from 2015 to the present. Its high-frequency revision offers a good foundation for generating annual water maps. Considering that water extent changes over time, a single-temporal DW image cannot adequately represent the annual extent of rivers. Thus, all DW time series images within one year were used to produce annual water maps. As reported by Venter et al. (2022), the mode algorithm can

generate the most common land cover for each pixel during the year, which was adopted in this study. The specific principle is illustrated in Fig. 4.

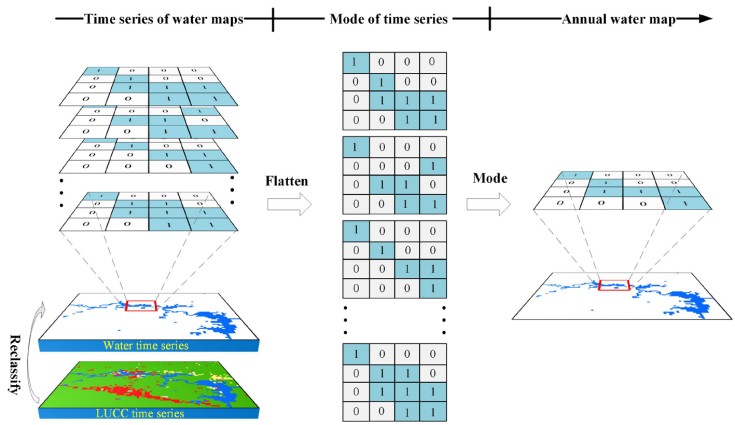

**Fig. 4. Illustration of annual water generation**

First, water images were generated by reclassifying the DW images. For an individual pixel of a water image, 1 represents

water, and 0 represents non-water. Then, water time series images were generated. Finally, the mode algorithm was used to aggregate the water time series images within one year into annual water maps. For example, if a pixel has time series values of (1, 0, 0, 1, 1), then the aggregated value of this pixel is 1. If a pixel has time series values of (1, 0, 0, 0, 1), then the aggregated

value of this pixel is 0. In this way, annual water maps were created via the mode algorithm. Additionally, considering the scarcity of DW images before 2016, yearly water maps for 2016 were produced via DW time series images from 2015-2016.

The above processing was implemented in the green tiles of Fig. 3.

### 3.1.2 Water extraction based on the EGLC dataset

For the data gaps in the DW images from 2017 to 2023, we used the EGLC dataset as a supplemental dataset. The EGLC dataset is an annual-scale global land use dataset that records water and eight other land use types. It is produced annually on the basis of Sentinel-2 time series images from 2017 to 2023. In this study, yearly water maps were directly produced via

reclassification of the EGLC dataset. The above processing was implemented in the blue tiles of Fig. 3.

### 3.1.3 Water extraction from Sentinel-2 images

Owing to the lack of 10-m LUCC or water datasets in 2016, we employed Sentinel-2 images to extract water data. Considering the scarcity of Sentinel-2 images before 2016, we collected all the images from June 2015 to December 2016. We classified water areas via the multiple index water detection rule (MIWDR) in the GEE platform, which was developed by

Deng et al. (2019). In the MIWDR, five spectral indices were calculated, including the modified normalized difference water index (MNDWI), normalized difference vegetation index (NDWI), enhanced vegetation index (EVI), and automated water extraction index for shadows and no shadows ($AWEI_{sh}$ and $AWEI_{nsh}$, respectively). Water areas were extracted via the criteria "$AWEI_{sh} - AWEI_{nsh} > -0.1$" and "MNDWI>EVI or MNDWI>NDVI". In this way, water time series from 2015 to 2016 were produced, which were then aggregated into annual water maps for 2016 via the mode algorithm. The above processing was

implemented in the light red tiles of Fig. 3.

### 3.2 Object-based hierarchical decision tree algorithm

To extract rivers from water areas, an object-based hierarchical decision tree algorithm was developed on the basis of geometric features and auxiliary datasets. To calculate the geometric features, chessboard segmentation was implemented in eCognition software, with scales larger than those of the columns and rows of the input water image. Five geometric features—

compactness, length/width, roundness, rectangular fit, and area—were calculated for the water objects. To analyse the geometric differences in water cover (e.g., rivers, lakes, and reservoirs), training samples were created on the basis of the 2020 CNLUCC map. Specifically, polygons of rivers, lakes and reservoirs were generated on the basis of the CNLUCC map. Random samples of each water cover type were created via a random sampling method. Then, the segmented water objects were overlaid with the training objects, and training objects of corresponding water types were obtained. Fig. 5 displays the

geometric features of rivers, lakes and reservoirs.

**Fig. 5. Violin plots and quartile box plots of geometric features. The columns represent three area range groups of 0–1000 ha, 1000–5000 ha, and >5000 ha, respectively. The rows represent the statistics of compactness, length/width, roundness and rectangular fit, respectively.**

The geometric statistics indicated that the geometry of the lakes and reservoirs clearly differed from the rivers. Specifically, the compactness, roundness and length/width of rivers were greater than those of lakes and reservoirs, whereas

the rectangular fit of rivers was less than that of lakes and reservoirs. Moreover, the above characteristics for the three water body types were more pronounced for water objects with larger areas. To reduce omission errors, we constructed five weak rules, namely, "compactness >2.3", "rectangular fit <0.5", "length/width >1.8" and "compactness >5.0". To reduce commission



errors, we used the five weak rules to construct strong rules. On the basis of our trials and errors, the rule set for river extraction was "compactness >2.3 & rectangular fit <0.5 & length/width >1.8" or "compactness >5.0".

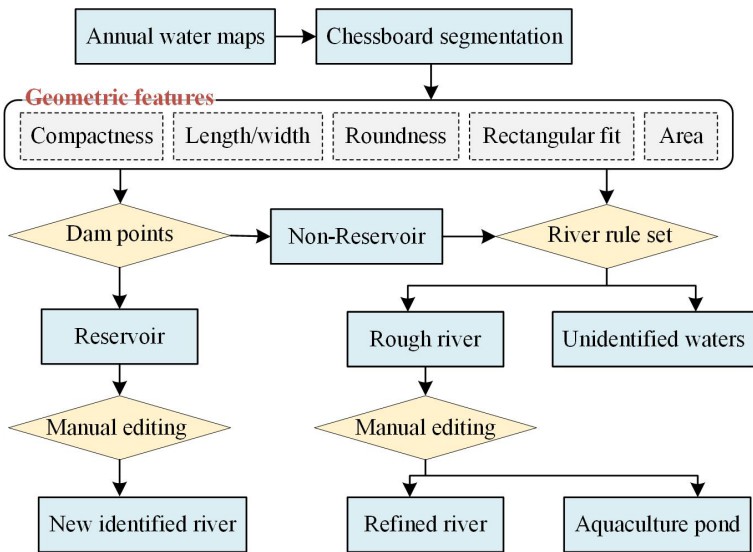

**Fig. 6. Workflow of the object-based hierarchical decision tree**

On the basis of the designed rule sets and auxiliary datasets, an object-based hierarchical decision tree (HDT) was developed to subsequently separate water cover types (Fig. 6). First, the dam points, including the Gaode reservoir POI, GOODD and OSM_Dam points, were used to identify reservoirs. These points were overlaid with water objects, and the selected water objects were labelled reservoirs. Then, the rivers were extracted via the river rule set. The remaining water objects were labelled as unidentified water. Finally, some manual edits were needed to improve the quality of the river maps, which involved three main situations: (1) Some aquaculture ponds were spatially connected with rivers, which cannot be separated via the river rule set. (2) The identified reservoirs may contain some rivers due to position errors at dam points. (3) Small rivers with narrow channels cannot be well identified via the river rule set because of their spatial disconnection. We manually revised these mapping errors.

**3.3 Accuracy validation**

To assess the mapping accuracy of rivers, test samples were produced by combining stratified random sampling and visual interpretation. First, river and non-river samples were produced via overlay, buffer and random sampling operations. Specifically, the rivers in the CNLUCC dataset for 2020 were extracted and then overlaid with the rivers from 2020. In the union regions, river samples were created via random sampling. After that, non-river samples were produced within a 300 m outside buffer of the union regions. Second, all random samples were visually interpreted by combining Collect Earth (CE), Google Earth (GE) and the GEE platform (Peng et al., 2024b). The CE software enables user-friendly sample management,



the GE provides high spatial resolution images, and the GEE offers median-composited Sentinel-2 images. Using these three

platforms, river and non-river samples from 2016 to 2023 were produced.

Finally, the accuracy indices, including the overall accuracy (OA), production accuracy (PA), user accuracy (UA) and F1

score, were calculated on the basis of the test samples. The F1 score measures the comprehensive accuracy of individual types

that balances the PA and UA, and is calculated as follows:

$$F1 = 2\frac{PA \times UA}{(PA + UA)} \times 100\% \tag{1}$$

### 3.4 Dataset intercomparison

To further validate the reliability of our river dataset, three existing river-related datasets—EA_Wetlands, CWaC and

CNLUCC—were used for intercomparison. Specifically, the rivers of EA_Wetlands in 2021, the rivers of CWaC in 2020, and

the rivers of the CNLUCC dataset in 2020 were used to test our river maps in the corresponding years. The first two datasets

have the same spatial resolution as our river dataset and were compared in 2021 and 2020. CNLUCC 2020 has a 30 m spatial

resolution with an OA of 91.2%, which was also used for data intercomparison because of its high mapping accuracy. To

comprehensively reflect their agreement, we aggregated the river extents within the spatial grid of 0.05°×0.05° and obtained

scatterplots and linear regressions with correlation coefficients (R) and root mean square errors (RMSEs).

### 3.5 Spatial–temporal analysis of China's areas

On the basis of our river maps from 2016 to 2023, the spatial patterns and areas of rivers at both the national scale and

river basin scale were analysed. Furthermore, the Theil–Sen method was used to calculate the change slope of river areas

within a grid of 0.05°×0.05°. This method is a robust nonparametric algorithm that can reduce the impact of outliers on the

computed slope (Qiu et al., 2024). Its formula is as follows:

$$S = Median(\frac{x_j - x_i}{j - i}) \quad \forall j > i \tag{2}$$

where S denotes the Theil–Sen slope, i and j denote years, and $x_i$ and $x_j$ denote river areas within a grid of 0.05°×0.05°

in different years. If $S < 0$, there was a decreasing trend in the river area of the grid. If $S = 0$, it indicates a stable river area. If

$S > 0$, there was an increasing trend in the river area of the grid.

## 4 Results

### 4.1 River accuracy assessments

Combining the DW dataset, GLEC dataset and Sentinel-2 images, we produced annual China river extent maps (CRED)





from 2016 to 2023. We validated the CRED via test samples that were manually inspected via visual interpretation. The CRED

achieved stable and good accuracy, with all OAs exceeding 96.8% each year (Table 2). For rivers, the UAs and PAs from 2016

to 2023 exceeded 95.3% and 91.3%, respectively. For the non-river areas, the UAs and PAs in individual years exceeded 96.0%

and 97.7%, respectively. The above results indicated that the omission and commission errors of our river maps were very low.

**Table 2. Accuracy assessments of our river maps from 2016 to 2023**

|  | River | | | Non-river | | | OA |
|---|---|---|---|---|---|---|---|
|  | UA | PA | F1 score | UA | PA | F1 score |  |
| 2016 | 96.8% | 92.9% | 94.8% | 97.9% | 99.1% | 98.5% | 97.6% |
| 2017 | 97.9% | 94.5% | 96.2% | 97.4% | 99.0% | 98.2% | 97.5% |
| 2018 | 96.2% | 91.3% | 93.7% | 96.0% | 98.3% | 97.1% | 96.0% |
| 2019 | 97.2% | 94.5% | 95.8% | 97.2% | 98.6% | 97.9% | 97.2% |
| 2020 | 96.5% | 94.2% | 95.4% | 96.9% | 98.2% | 97.5% | 96.8% |
| 2021 | 95.4% | 94.4% | 94.9% | 97.5% | 98.0% | 97.7% | 96.8% |
| 2022 | 95.3% | 94.9% | 95.1% | 97.5% | 97.7% | 97.6% | 96.8% |
| 2023 | 98.4% | 96.5% | 97.5% | 98.5% | 99.3% | 98.9% | 98.5% |

In addition, the F1 score, which measures the comprehensive accuracy of individual types, was calculated. The results

indicated that the F1 scores of rivers and non-rivers were greater than 93.7% and 97.1%, respectively. On the basis of the above

analysis, we believe that the rivers in the CECD from 2016 to 2023 achieved reliable accuracy.

**4.2 Intercomparison with river-related datasets**

To further test the reliability of the CRED, three river-related river datasets, including CNLUCC, CWaC and

EA_Wetlands, were chosen for data intercomparison. To quantify the consistency among datasets, river fractions within a 0.05°

by 0.05° grid were counted (Fig. 7). The results indicated that our CRED exhibited high correlation coefficients (R) with the

three river-related datasets. CNLUCC, considered one of the most reliable land use datasets, had the highest R among the three

datasets, with a value of 0.832. The CNLUCC was produced by experienced interpreters through visual interpretation on the

basis of Landsat images. The high R value between the CNLUCC dataset and the CRED indicated the good reliability of the

rivers in the CRED. CWaC and EA_Wetlands were generated via intelligent algorithms at a 10 m spatial resolution, and their

R values with the CRED were 0.775 and 0.750, respectively. Overall, our river maps exhibited good consistency with existing

river-related datasets.





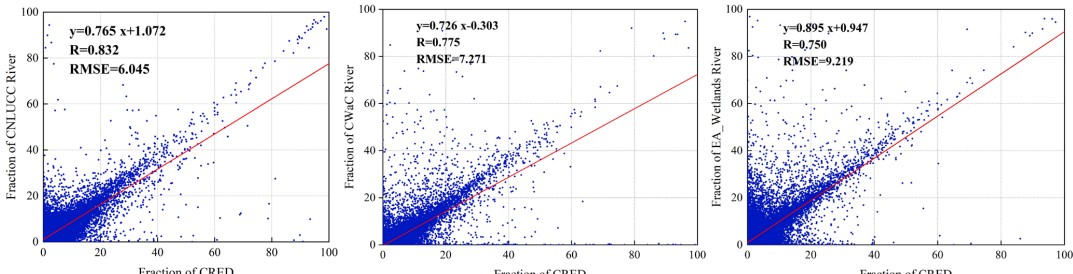

**Fig. 7. Scatterplots of the river fraction between the CRED and CNLUCC, CWaC and EA_Wetlands. The river fraction was aggregated within 0.5° by 0.5° grids.**

To inspect the spatial consistency between the CRED and the three river-related datasets, the spatial distributions of Chinese rivers in the corresponding datasets were mapped (Fig. 8). These river maps generally shared similar distributions. Among the three water-related datasets, CNLUCC had the most similar patterns with the CRED (Fig. 8 (c) & (e)). For the EW_Wetlands and CWaC, their rivers had considerable differences from the CRED. The EA_Wetlands dataset delineated more rivers west of the Continental Basin (CB) and southwest of the Songhua and Liaohe River Basin (SLRB) (Fig. 8 (a)). CWaC misclassified many small-area patches as rivers, such as ice, snow, and shadows (Fig. 8 (d)). In addition, river areas of individual river maps were calculated, and six typical regions were selected to check spatial details among river datasets (Fig. 8 (f)).

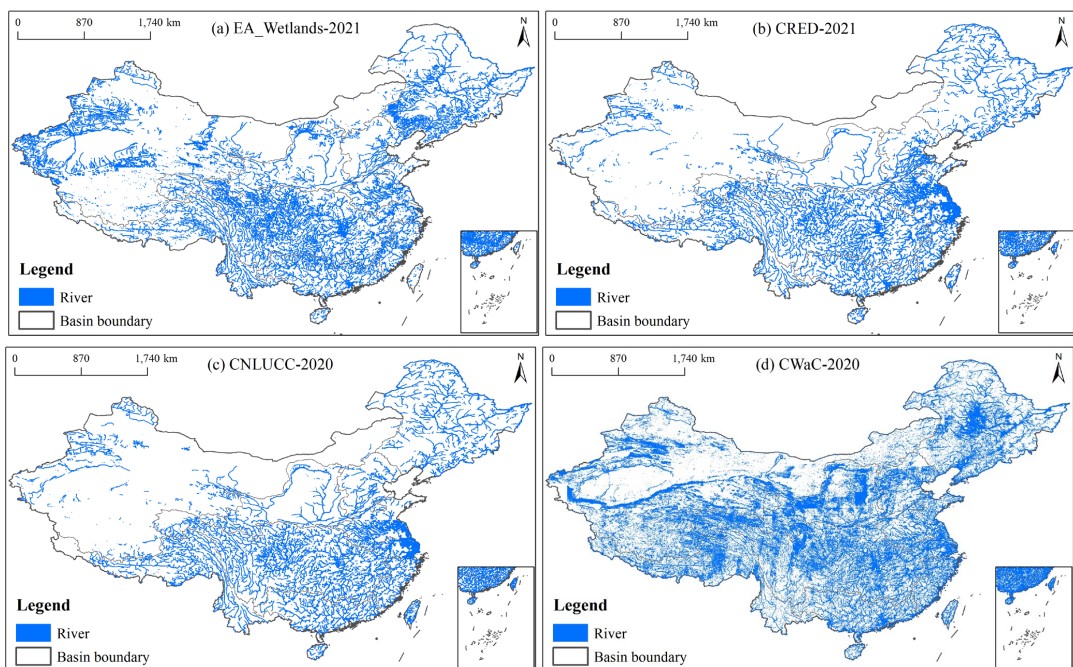

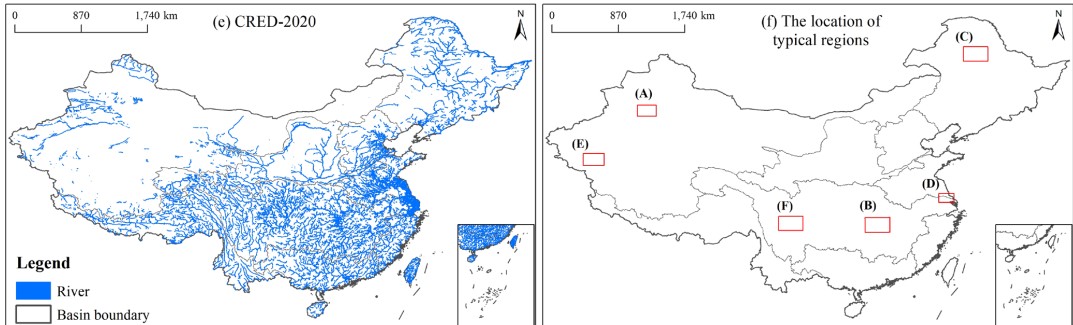

**Fig. 8. River patterns of EA_wetlands (2021), CLUCC (2020), CWaC (2020) and the CRED (2020/2021). (f) displays six typical regions, and their rivers are shown in the supplementary materials.**

For EA_Wetlands and the CRED in 2021, the river areas were 64886.84 km$^2$ and 47397.75 km$^2$, respectively. The river

area in EA_Wetlands was larger than that in the CRED, mainly because of various factors. On the one hand, the EA_Wetlands

dataset better represented rivers in complex mountainous areas and seasonal rivers than the CRED (Fig. S2(A)). On the other

hand, the EA_Wetlands dataset misclassified some linear water areas as rivers, such as Dongting Lake (Fig. S2(B)).

The river areas of the CNLUCC dataset and the CRED in 2020 were 47563.20 km$^2$ and 44694.64 km$^2$, respectively. The

river areas of the two datasets were nearly equal. However, due to spatial resolution differences in satellite images, some

differences were still identified. CNLUCC did not effectively extract narrow rivers in mountainous areas (Fig. S3(C)) or

identify canals/channels along coastal regions in Southeast China (Fig. S3(D)). However, the CRED was able to effectively

recognize these rivers.

The river areas of CWaC and the CRED in 2020 were 33877.90 km$^2$ and 44694.64 km$^2$, respectively. The river area in

CWaC was smaller than that in the CRED, likely due to an underestimation of river extents by CWaC. CWaC was produced

by combining Sentinel-1 SAR (synthetic aperture radar) images and Sentinel-2 MSI (multispectral image) images. As reported

by Li and Niu (2022), water bodies extracted via SAR images are often underestimated. Consequently, the river extents mapped

by CWaC may also be underestimated. In addition, considerable small patches, such as ice, snow and shadows, were

misclassified as rivers in CWaC (Fig. S4 (E)). Moreover, the CRED was superior to CWaC in terms of data postprocessing.

For example, CWaC misclassified some reservoirs as rivers, whereas the CRED accurately excluded them (Figure S4 (F)).

**4.3 Spatial pattern and area changes of China's rivers**

In this study, the annual river extents of Chinese rivers from 2016 to 2023 were mapped, and their spatial distributions

are displayed in Fig. S5. China's rivers are characterized by more southern rivers and fewer northern rivers, more eastern rivers

and fewer western rivers. Generally, most rivers are distributed in Southeast China and Southwest China. Northeast China also

has many rivers.

In 2023, the river areas of China were 44,948.78 km$^2$. Among the nine river basins, the YaRB had the largest river area



and densest river network, accounting for 42.55% of China's river area in 2023 (Fig. 9). The others were the SLRB, PRB, HuRB, YeRB, SWB, SEB, CB, and HaRB in descending order, with river areas accounting for 14.10%, 11.81%, 7.40%, 7.21%, 6.33%, 5.24%, 3.17% and 2.00%, respectively. The CB is the largest river basin in China but has the sparest river network. The HaRB is the smallest river basin in China and has the smallest river areas; however, it has a relatively dense river network.

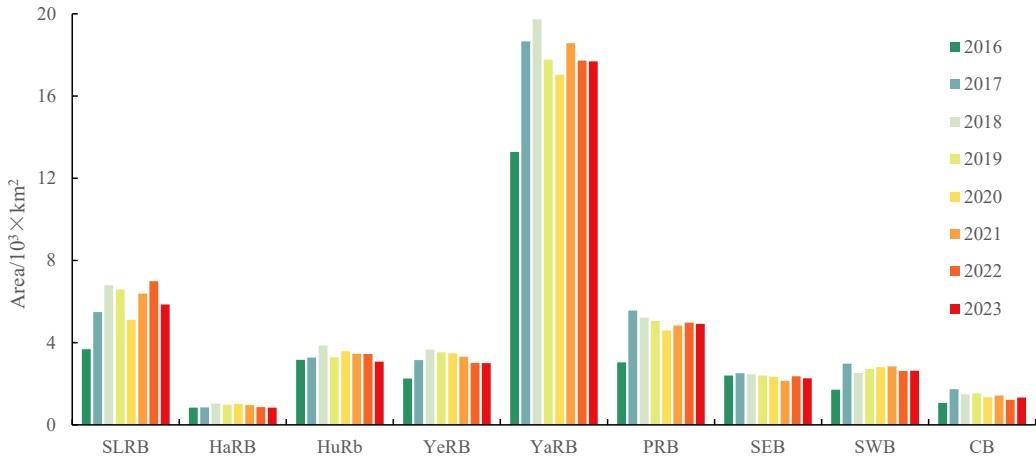


**Fig. 9. River areas from 2016 to 2023 in nine river basins**

Temporally, the river areas of China from 2016 to 2023 were characterized by an initial increase, followed by a decrease and then a slight increase. This change trend was also observed for the SLRB, HuRB, YeRB, PRB, SEB, SWB, and CB. For the HaRB and YeRB, the river area first increased but then decreased. In the past eight years, the river areas of 2018 in the

SLRB, HaRB, HuRB, YeRB and YaRB were the largest, whereas the river areas of 2017 in the PRB, SEB, SWB and CB were the largest.

### 4.4 Changing characteristics of China's rivers

To explicitly examine the change characteristics of China's rivers, the Theil-Sen slope of river areas within 0.05°×0.05° grids was calculated (Fig. 10). More than half of China's rivers had a negative slope, indicating a decreasing trend in river

area. The decreased areas were distributed mainly in coastal regions, such as the Yangtze River Delta, HaRB, HuRB, SEB and PRB. Most of these regions had slope values ranging from 0 to 1 (Fig. 10 (b)). These trends and distribution characteristics were related mainly to human activities. Regions with decreased river areas experienced high-level economic development. Correspondingly, their human disturbances were more frequent. Over the past eight years, the expansion of construction land, industrial and agricultural development, and other areas in these regions have contributed to the degradation of river areas.

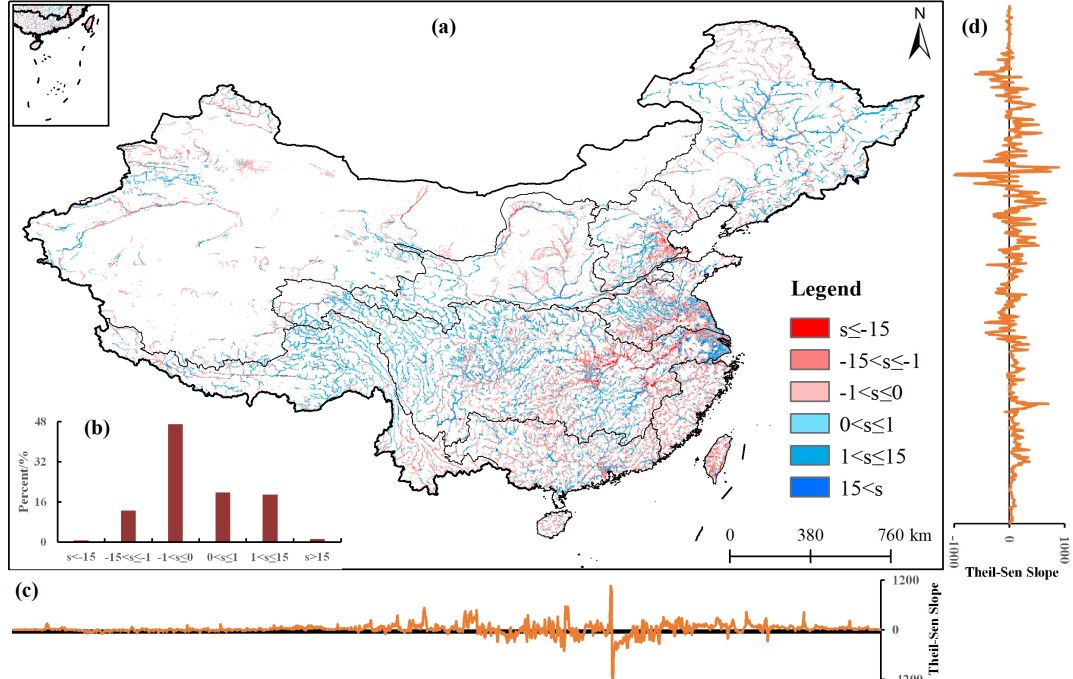


**Fig. 10 Theil-Sen slope of rivers in the CRED. (a) Theil–Sen slope for 0.05°×0.05° grids in China. (b) Histogram of Theil–Sen slopes for the river grids. (c) and (d) Slope histograms with 0.05° increments in longitude and latitude.**

For grids with increasing river areas, most are located in the southern SLRB, western YaRB, and southeastern CB. Most of the increased river grids had slope values ranging from 0 to 15, which accounted for 38.70% of the total river grids (Fig. 10

(a&b)). For increased river grids, their change trends and distributions could be linked with ecological restoration and water conservancy projects. On the one hand, local governments have implemented a series of policies to protect natural ecosystems. Large encroachments on river ecosystems are strictly prohibited in China. On the other hand, some water conservancy projects, such as the Yellow River Diversion Irrigation project and the South-to-North Water Diversion project, have greatly promoted the increase in river area.

In addition, we also calculated the river change slope by longitude and latitude at a 0.05° interval (Fig. 10 (c&d)). From the perspective of longitude, rivers with significant changes were mainly concentrated in central and eastern China, covering the southern coastal regions. These changes were most related to high-speed economic development. From the perspective of latitude, rivers with significant changes were distributed mainly in northern China. These change characteristics were linked mainly to the new constructions of cannels and conservancy facilities.



## 5 Discussion

### 5.1 Reliability of river mapping

In this study, we successfully produced annual river maps of China from 2016 to 2023. To our knowledge, this is the first comprehensive effort to map annual river extents across China at a 10 m spatial resolution. The successful implementation of our study can be attributed to three factors: the availability of multisource LUCC datasets, the reliability of existing water extraction algorithms, and the effectiveness of our river mapping algorithm.

First, multisource LUCC datasets at a spatial resolution of 10 m provide a good foundation for river mapping, such as ESRI's world cover (Zanaga et al., 2021), FROM-GLC10 (Gong et al., 2019), DW, and EGLC. The DW dataset, with a revisit frequency of 2–5 days, allowed us to capture seasonal variations in water extent (Brown et al., 2022). However, it is often unavailable in Southwest China and the Tibetan Plateau because the DW dataset was produced using only Sentinel-2 images with less than 35% cloud coverage. To address this limitation, the EGLC was used as supplementary data in areas where the DW dataset was missing, ensuring more comprehensive river mapping.

Second, existing water extraction algorithms also support river mapping. Owing to the scarcity of Sentinel-2 images from 2015 to 2016, the EGLC dataset was unavailable in 2016, and significant portions of the DW dataset were also missing for China during this period (Karra et al., 2021; Wang et al., 2023c). To address this issue, the MIWDR algorithm was employed to extract water areas in 2016, which served as supplementary data for the DW dataset. The MIWDR, which is designed with multiple water indices, has been shown to be effective at extracting water from mountainous and urban areas, making it highly suitable for large-scale water classification (Deng et al., 2019).

Third, we developed an object-based hierarchical decision tree algorithm for river extraction. Rivers, characterized by their linear and narrow shapes, have distinct geometries compared with other water covers (e.g., lakes and reservoirs) (Peng et al., 2023). To effectively extract rivers, five geometric features, which were calculated via chessboard segmentation, were used to develop a hierarchical decision tree. Compared with pixel-based methods, our algorithm can handle the spectral similarity of water cover types. Compared with general object-based methods, our algorithm has good robustness, as its rules and thresholds do not change over time or across regions.

An accuracy validation indicated that our annual river maps achieved high accuracy, with OAs exceeding 96%. For the individual accuracy metrics, the UAs, PAs and F1 scores of the river areas were greater than 95.3%, 91.3% and 93.7%, respectively. To further check the reliability of our CRED, three existing river datasets, including EA_Wetlands, CNLUCC, and CWaC, were selected for intercomparison. The results showed that our CRED exhibited similar spatial distributions to those of the existing datasets, with R values exceeding 0.75. Moreover, the CRED was superior in mapping small and narrow rivers and reducing river misclassifications. Overall, the above analysis confirmed that our river maps are both accurate and reliable, effectively delineating river extents and dynamics.



**5.2 Improvements and potential contributions of the CRED**

Owing to their ecological and economic importance, river datasets have been produced to support river management and economic development. These datasets can be divided into river line datasets and river polygon datasets (Table 3). River line datasets record channel lines and reflect river spatial distributions well. However, these datasets cannot delineate river extents 355 and area changes. River polygon datasets delineate river extents with polygon patches. However, most of these datasets are limited by low temporal and spatial resolutions. On the one hand, long-term river polygon datasets are scarce and fail to reflect river dynamics. On the other hand, the spatial resolutions of these datasets are intermediate or low and are thus insufficient to delineate river extents in detail. In contrast, the CRED has better spatial and temporal resolution. It maps annual river extents in China at a spatial resolution of 10 m, which can monitor river extents and their dynamics well.

**Table 3. List of existing river datasets covering the globe or China. The dashed lines indicate that the corresponding information is unavailable.**

| Type | Name | Spatial extent | Spatial resolution | Time period | Resource |
|---|---|---|---|---|---|
| River lines | GRRATS | Global | --- | --- | (Coss et al., 2020) |
| | HydroRIVERS | Global | 500 m | --- | (Lehner and Grill, 2013) |
| | MCRW | China | 30 m | 1990–2015 | (Yang et al., 2020a) |
| | Reach-level Bankfull river | Global | 30 m | --- | (Lin et al., 2020) |
| River polygons | GLWD | Global | 1 km | 1980s | (Lehner and Doll, 2004) |
| | GRWL | Global | 30 m | --- | (Allen and Pavelsky, 2018) |
| | EA_Wetlands | East Asia | 10 m | 2021 | (Wang et al., 2023a) |
| | CAS_Wetlands | China | 30 m | 2015 | (Mao et al., 2020) |
| | CWaC | China | 10 m | 2020 | (Li and Niu, 2022) |
| | River_OSM | Global | --- | --- | https://download.geofabrik.de/asia/china.html |

Long-term and accurate river extent datasets significantly benefit biodiversity conservation and sustainable river ecosystem management (Gonzalez-Ferreras and Barquin, 2017). For example, river ecosystems not only provide habits and 365 food for fish, amphibians, birds and aquatic plants but also serve as curial natural corridors for species migration, connecting



different ecological zones (Mar Sanchez-Montoya et al., 2022). The CRED maps, with detailed spatial information and dynamic changes, can be used as baseline data for managing, protecting and improving river ecosystems. On the basis of the CRED, many river-related indices, such as hydrological connectivity, patch area, and land degradation, can be calculated, providing valuable insights into the environmental health of river ecosystems (Yan and Niu, 2019).

Moreover, the CRED can also effectively serve economic development. A river equipped with good shipping capacity can greatly facilitate commodity transportation and trade ((Spear et al., 2024)). In southeastern China, for example, the dense river network provides good access to shipping, which largely contributes to local economic development ((Liu et al., 2018a)). The CRED records annual river extents over the past eight years and has good potential for enhancing shipping capacity. Shipping conditions, including the river width, area and spatial connection, can be easily derived from the CRED. In agriculture,

rivers provide irrigation water for arable land, ensuring crop growth (Lu et al., 2023). Regular and dynamic monitoring of river distribution and area changes is beneficial for agricultural production and management.

### 5.3 Uncertainties, limitations and improvements

This study focused on river extraction from China and successfully produced high-quality river maps from 2016 to 2023. However, there are still some uncertainties and limitations. First, the multisource LUCC datasets and water extraction

algorithm used for CRED production may result in uncertainties. As previously mentioned, the DW dataset cannot cover all of China. To address this issue, the EGLC dataset and water classification by the MIWDR algorithm were used as supplementary datasets. The three datasets shared different characteristics in terms of water mapping. The DW well captured water seasonal variation but missed many water areas in mountain areas (Li and Wang, 2024). The EGLC delineates only annual water extents without seasonal information (Venter et al., 2022). The MIWDR algorithm misclassified some ice, snow

and shadows as water. The above characteristics of the three datasets may lead to errors in river extraction.

In addition, the DW, EGLC, and MIWDR methods failed to adequately map water areas on both the Tibetan Plateau and in Southwest China. These errors were related mainly to mountain shadows and complex environments. Water areas share spectral characteristics similar to those of mountain shadows, making them prone to misclassification (Wang et al., 2023b). Complex environments, such as cloud contamination and ice/snow, make accurate extraction of water difficult. In addition,

the mode algorithm, which is used to map the most common rivers on an annual scale (Calderon-Loor et al., 2021), may also result in uncertainties. Owing to seasonal variability, the boundaries of mapped rivers may be contentious. How to define the extent of a river, such as by using the maximum, minimum or medium water inundated areas, is a debated issue.

In future studies, the data quality and time range of the CRED could be further improved. As previously discussed, many uncertainties in the CRED are largely caused by multisource LUCC datasets or water datasets. Despite the availability of water

datasets at 10 m spatial resolution(Li et al., 2023; Vanderhoof et al., 2023; Yang et al., 2020b), long-term global water datasets

are still scarce. Thus, future attempts should focus on developing robust and accurate water classification algorithms for Sentinel-1/2 images, which are aimed at producing long-term and large-scale water data with high consistency. In return, these water classification studies support river extraction well. Additionally, as Sentinel-1/2 imagery accumulates, the CRED should be updated in a timely manner to extend its time range.

**6 Data availability**

The China River Extent Dataset (CRED) is publicly available for download from the Zenodo repository https://doi.org/10.5281/zenodo.13841910 (Peng et al., 2024a). It delineates river extents across China from 2016 to 2023. Correspondingly, the CRED contains eight shapefiles. It has an area attribute, with units of hectares (ha). The projection of the CRED is WGS_1984_Albert.

**7 Conclusion**

Rivers play crucial roles in terrestrial ecosystems and human well-being, not only by providing freshwater resources and supporting biodiversity but also by serving the functions of navigation, cultural recreation and flood disaster prevention. To support the high-quality development of China, it is vital to produce sequential and fine-resolution river extent maps. On the basis of a literature review, however, fine-resolution annual river extent maps across China have rarely been investigated in

past studies. Thus, to better understand the spatial distribution and dynamics of China's rivers, this study generated the first Sentinel-derived annual China river extent dataset (CRED) from 2016 to 2023 on the basis of the DW dataset, EGLC dataset and MIWDR algorithm.

An accuracy evaluation indicated that the OAs of the CRED from 2016 to 2023 exceeded 96.0%. For the individual accuracy indices, the UAs, PAs and F1 scores of the rivers exceeded 95.3%, 91.3% and 93.7%, respectively. To further check

the reliability of our CRED, data comparisons with EA_Wetlands, CNLUCC and CWaC were implemented. The results indicated that the CRED shared similar distributions with the three datasets, with R values over 0.75. Moreover, the CRED dataset was superior to existing river-related datasets in terms of spatial details, river misclassifications and omissions. Spatial–temporal analysis revealed that rivers in China are more abundant in East China than in West China and more abundant in South China than in North China. The YaRB had the largest river area among the nine river basins, accounting for 42.55% of

China's rivers. From 2016 to 2023, the areas with decreases in rivers were mostly located in Southeast China, whereas the areas with increases in rivers were mostly distributed in Central and Northeast China. Overall, this study produced high-quality annual river maps for China and explored their dynamics from 2016 to 2023, which is important for the protection, management and sustainable use of rivers.

**Author Contributions**

FP conceived this study. The methodology design, manuscript wiring, and format analysis were done by FP and XW. Data processing, software, and programming were done by FP, SB and XW. FP and WJ supervised, acquired and administered the project. Formal analysis and manuscript improvement were done by MM. Quality control and accuracy validation were done by FP, SB, WJ, MM, and XW.

**Competing interests**

The contact author has declared that none of the authors has any competing interests

**Acknowledgements**

The authors greatly appreciate the free access to Sentinel-2 images provided by the European Space Agency (ESA), the dynamic world (DW) dataset provided by Google in partnership with the National Geographic Society and the World Resources Institute, the ESRI global land cover (EGLC) dataset provided by ESRI, the China land use/cover change (CNLUCC)

dataset provided by the Resource and Environmental Science Data Platform, the EA_Wetlands dataset provided by the Northeast Institute of Geography and Agroecology, Chinese Academy of Science, the China water cover (CWaC) map provided by the Aerospace Information Research Institute, Chinese Academy of Science, the reservoir point of interest (POI) dataset provided by Amap, and the OSM_Dam dataset and the GOODD dataset provided by Global Dam Watch. We also express our gratitude to the Google Earth Engine platform, which made it convenient to access and process Sentinel-2 images, the DW

dataset and the EGLC dataset.

**Financial Support**

This research was jointly supported by the National Natural Science Foundation of China (grant no. 42301413), the Open Fund of the State Key Laboratory of Remote Sensing Science (grant no. OFSLRSS202426) and the National Natural Science Foundation of China (grant no. U21A2022).

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
