# Peer review of "Annual river dataset in China: a new product with a 10 m spatial resolution from 2016 to 2023"

_Earth System Science Data, 2024_

## Referee Comment (RC1)

The manuscript introduces a multi-temporal China annual river extraction framework, which includes a multi-data source-based water extraction module and an object-based hierarchical decision tree river extraction algorithm, and produces annual China river extent maps (CRED) from 2016 to 2023. However, the paper needs further improvement in terms of its structure and readiness for publication. The motivation and innovation of the research should be clarified.

Some major comments are as follows.
(1) The motivation for using multisource datasets for water extraction should be better explained. The authors state that the choice of data sources (DW, EGLC, and Sentinel-2) is based on their availability, in that order. However, the river mapping results for China in 2016, primarily using Sentinel-2, show no significant differences compared to other years. Is the proposed method aimed at achieving higher extraction efficiency, or is it designed to enhance accuracy?
(2) In the proposed approach, the geometric rules for river extraction were based on the 2020 CNLUCC map. As shown in Fig. 8, the extraction results from CRED exhibit significantly higher spatial consistency with the CNLUCC map compared to the other two comparison datasets. Did the authors consider using different datasets during the geometric rule extraction or the result comparison process?
(3) In the statistical results for river areas from 2016 to 2023, the river area in 2016 was noticeably smaller than in other years. Was this phenomenon also observed in non-river water bodies? It would be helpful to include the accuracy of water extraction for each year.

More specific comments are as follows.
(1) Sensitivity analysis is required to validate the feasibility of the proposed method for extracting water body extents using different data sources across different tiles/periods.
(2) The water extraction section in Figure 2 could be clearer. Presenting data for all years together to generate the water time series may cause confusion and fails to adequately convey the meaning of "For areas where DW observations were missing" in line 92.
(3) In line 199, please clarify what "the rivers from 2020" refers to. If it refers to the extraction results from this study for 2020, please clarify the potential impact of generating validation samples based on extraction results on the randomness and representativeness of the samples.
(4) The sample size for validation is unclear. Please provide details on the distribution and quantity of the validation samples in Section 4.1.
(5) The resolutions of the three existing products used for comparing river extraction results are not exactly the same. Did the authors perform any resampling or other processing when comparing river areas to eliminate the area differences caused by resolution?

(6) In line 270, it is mentioned that the CRED dataset outperforms the existing most accurate products in extracting narrow rivers in mountainous areas. Did the authors consider providing a more precise definition of narrow rivers to highlight the advantages of this product?

(7) The area difference mentioned in line 273 between the river areas of CWaC and the CRED in 2020 is inconsistent with the visualization results in Fig. 8. Please provide more detailed comparisons of the water bodies extracted result.

---

## Author Comment (AC1)

**Response to The Comments from Reviewer #1**

The manuscript introduces a multi-temporal China annual river extraction framework, which includes a multi-data source-based water extraction module and an object-based hierarchical decision tree river extraction algorithm, and produces annual China river extent maps (CRED) from 2016 to 2023. However, the paper needs further improvement in terms of its structure and readiness for publication. The motivation and innovation of the research should be clarified.

**Response:**

Thank you for taking your precious time and making diligent efforts to review our manuscript. The valuable comments and constructive suggestions are definitely helpful, and we sincerely appreciate them for improving our paper. We have carefully studied the comments and revised the manuscript point-by-point. All modifications have been marked in the revised manuscript. One again, thank you for your valuable comments.

**Some major comments are as follows.**

**1.** The motivation for using multisource datasets for water extraction should be better explained. The authors state that the choice of data sources (DW, EGLC, and Sentinel-2) is based on their availability, in that order. However, the river mapping results for China in 2016, primarily using Sentinel-2, show no significant differences compared to other years. Is the proposed method aimed at achieving higher extraction efficiency, or is it designed to enhance accuracy?

**Response 1**:

Thanks for your valuable comments. It is helpful to enhance our manuscript. We have made revisions and explanations for the motivation of using multisource datasets (DW, EGLC and Sentinel-2) in the manuscript.

> 90    areas had high mapping accuracy. Statistical analysis revealed that many regions in China had no or few good observations (Fig. S1). This occurred because the DW images were produced from Sentinel-2 images with less than 35% cloud coverage. Sentinel-2 images used in DW dataset may have could contaminations, which also resulted in missing observations. Thus, the complete coverage of the entire China to produce a complete China river map cannot be achieved by using DW dataset alone.
>
>          To facilitate image computation, we divided China into 52 tiles using a 5°×5° grid (Fig. S1). If a tile contains considerable
> 95    pixels with fewer than 3 valid observations, it was defined as a data insufficient tile. For these tiles, we used the ESRI global land cover (EGLC) dataset to supplement the data (Karra et al., 2021). The EGLC dataset was produced on the basis of

The Dynamic World (DW) was a 10-m spatial resolution land use dataset from 2015 to the present, with a revision of 3-5 days. The high revisit frequency of DW allows it to effectively capture the seasonal variations of water bodies. Therefore, DW was chosen as the primary dataset for river classification. However, since the DW use only images with cloud coverage below 35% for classification, it exhibits significant data gaps in considerable regions of China. As shown in Figure T1, the value of each pixel represents the number of valid observations within a year, excluding bad

observations affected by cloud contamination, cirrus, and cloud shadows. It is shown that there are significant data gaps in Southwest China in 2023, such as in tiles 14, 15 and 16. To address this issue, the EGLC was selected to substitute the DW dataset in these tiles with missing data, and were used to generate water maps from 2017 to 2023. However, the EGLC have no data in 2016.

[Figure]

Figure T1. Valid observations for individual pixels in DW image of 2023. This figure was also displayed in Supplementary material

To make the best and maximize the use of Sentinel-2 imagery and extend the temporal span by producing an additional year of river map, we utilized the Sentinel-2 imagery to substitute the DW datasets with considerable invalid or insufficient observations from 2015-2016. We applied the multiple index water detection rule (MIWDR) to Sentinel-2 imagery to generate water time series and composited them into annual water map for 2016 using mode algorithm. In summary, the temporal span of three datasets is illustrated as Figure T2.

[Figure]

Figure T2. Time span of three datasets

**2.** In the proposed approach, the geometric rules for river extraction were based on the 2020 CNLUCC map. As shown in Fig. 8, the extraction results from CRED exhibit significantly higher

spatial consistency with the CNLUCC map compared to the other two comparison datasets. Did the authors consider using different datasets during the geometric rule extraction or the result comparison process?

**Response 2:**

Thank you for your valuable comments. In our study, the CNLUCC were used to generate training samples. These samples were used to explore the geometric difference of water covers (e.g. river, lake and reservoir) and determine appropriate thresholds of each geometric features. Then, the rule set for river extraction was developed. The developed algorithm for river mapping is robust and effective. This algorithm is not supervised algorithm, and is independent of training samples. Its rules and thresholds were constant, did not change over time and regions.

The high consistency between our CRED and CNLUCC is mainly due to their high mapping accuracy of rivers. The CNLUCC was a 30-m dataset with detailed land use types, which was produced by human-computer interactive process. Its extensive manual interpretation and strict data production procedures ensure the high accuracy of the data. The CRED was produced using the accurate river mapping algorithm, and was further improved by post-processing operations. The CRED also achieved high accuracy of rivers. Thus, these two datasets had good consistency.

We did not use different datasets for the algorithm development. To further illustrate the accuracy, robustness and effectiveness of our algorithm, we implemented river mapping results using five data sources (i.e. DW, ELGC, Sentinel-2, ESRI, and JRC-GSW). The sensitive analysis was conducted in our manuscript. The detailed descriptions were added in sub-section 5.1.

**3.** In the statistical results for river areas from 2016 to 2023, the river area in 2016 was noticeably smaller than in other years. Was this phenomenon also observed in non-river water bodies? It would be helpful to include the accuracy of water extraction for each year.

**Response 3**:

Thanks for careful review and offering valuable comments. Indeed, the river area in 2016 was noticeable smaller than other years. This phenomenon was mainly attributed to two aspects. First, due to large gaps in DW datasets from 2015 to 2016, the Sentinel-2 image was used to extract waters using multiple index water detection rule (MIWDR) in 31 out of 52 tiles in China. However, due to its low observation frequency and the impact of cloud contaminations, the available Sentinel-2 images from 2015 to 2016 were relatively scare. This limitation may result in uncertainties for mapping river extents. Second, the MIWDR algorithm that applied to Sentinel-2 images exhibit different performance in term of water classification, compared with deep learning algorithm that adopted by DW and EGLC. Based on the sensitive analysis in sub-section 5.1, it was found that the MIWDR could well extract large and pure waters, while exhibited poor performance in seasonal

waters or mixed pixels of waters. This characteristics of MIWDR may also lead to underestimations of river extents. We discussed this uncertainty in our Discussion sections.

445   annual water extents without seasonal information (Venter et al., 2022). The MIWDR algorithm misclassified some ice, snow and shadows as water, and exhibited poor performances for seasonal waters or mixed pixels of waters. Meanwhile, due to the low observation frequency and the impact of cloud contaminations, the available Sentinel-2 images from 2015 to 2016 were relatively scare, which may result in underestimations of waters. These issues lead to the river area in 2016 was noticeable smaller than that in other years. In addition, the extent and completeness of manual corrections varied across different years,

**More specific comments are as follows.**

**1.** Sensitivity analysis is required to validate the feasibility of the proposed method for extracting water body extents using different data sources across different tiles/periods.

**Response 1**:

Thanks for your valuable comments. As suggested by the comment, we have added sensitivity analysis in our manuscript.

To evaluate the feasibility of our river mapping algorithm, five data sources were collected to implement river mapping. The DW, EGLC and waters derived from Sentinel-2 images using MIWDR were selected. Additionally, the ESRI WorldCover (ESRI) and JRC Global Surface Water (JRC-GSW) were also chosen to apply our algorithm for river extraction. We implemented river classifications using these five datasets for tile 21 in 2021. To explicitly illustrate the sensitive analysis, a new section has been added in the manuscript.

**5.1 Sensitivity analysis of our river extraction algorithms**

350    To further evaluate the feasibility of our river mapping algorithm, we used different datasets to extract rivers for tile 21 in 2021. On the one hand, the three datasets used in our study, namely DW, EGLC and waters derived from Sentinel-2 images using MIWDR (MIWDR), were selected to extract rivers. On the other hand, ESRI land use dataset and JRC-GSW water dataset were additionally selected to apply our algorithm for river extraction. The spatial distribution of rivers in five datasets were shown in Figure 11. Results indicated that rivers in different water maps could be effectively extracted. Specifically, our

355    algorithm could accurately extract the rivers shown in corresponding water maps.

[Figure]

**Figure 11. Spatial distribution of rivers and waters in different data sources. The tile number was 21 and the year was 2021.**

       However, due to differences of water extents in different datasets, the extracted river maps show significant variations.

360    The DW and EGLC adequately mapped yearly water extents. Many complex rivers (e.g. narrow rivers) were well displayed in these datasets. Thus, river networks in DW and EGLC were dense and complete (Figure 11 (a) & (b)). In contrast, rivers in MIWDR and ESRI were relatively spare, due to considerable narrow waters were not identified (Figure 11 (c) & (d)). For the JRC-GSW, the large rivers were effectively identified, while narrow rivers were not detected ((Figure 11 (e)). This was primarily due to the limitation of its 30 m spatial resolution. The above regulations were also validated in two typical regions

865 (Figure 12). The DW and EGLC mapped more waters, and narrow rivers in these datasets were well extracted. For the MIWDR, ESRI and JRC-GSW, large rivers could be accurately extracted, while considerable narrow rivers were missed.

[Figure]

**Figure 12. River distributions of different data sources in two typical regions. The satellite images (a1) and (b1) were Sentinel-2 images that composited using median algorithm based on time series images within 2021.**

870 To quantitatively evaluate the accuracy of rivers from different datasets, we produced 100 test samples using random sampling method and visual interpretation. Accuracy validation indicated that rivers in DW, ELGC and ESRI achieved good accuracy, with UAs and PAs exceeding 87%. For the MIWDR and JRC-GSW, the UAs of river classification were low, which indicated many rivers, most of narrow rivers, were misclassified or omitted. In contrast, the high PAs of rivers in MIWDR and JRC-GSW shown that rivers displayed in corresponding water maps can be accurately mapped. The above conclusions were

875 consistent with the spatial analysis of rivers in five datasets. It should be noted that the accuracy assessments in tile 21 were different with that of China's rivers. This mainly because the large rivers accounted for a significant proportion of rivers in China, and our algorithm was accurate and effective for rivers with large areas and spatial continuity.

**Table 3. Accuracy assessments of different datasets in 2021**

|  | DW | ELGC | MIWDR | ESRI | JRC-GSW |
|---|---|---|---|---|---|
| PA | 83.83% | 90.32% | 67.74% | 80.35% | 48.21% |
| UA | 91.16% | 87.5% | 95.45% | 95.74% | 96.43% |
| F-score | 87.34% | 88.89% | 79.24% | 87.37% | 64.28% |

**2.** The water extraction section in Figure 2 could be clearer. Presenting data for all years together to generate the water time series may cause confusion and fails to adequately convey the meaning of "For areas where DW observations were missing" in line 92.

**Response 2:**

Thanks for your careful review and offer useful suggestions. We have corrected the Figure 2 in the manuscript. The modified Figure is also shown below.

[Figure]

Figure T3. Workflow of annual river extraction. Three datasets marked by back boundaries was chose for river extraction. It should be noted that the boundaries of DW and EGLC varied across different years.

We have rewritten the sentence in line 92.

> To facilitate image computation, we divided China into 52 tiles using a 5°×5° grid (Fig. S1). If a tile contains considerable
>
> 95 pixels with fewer than 3 valid observations, it was defined as a data insufficient tile. For these tiles, we used the ESRI global
>
> land cover (EGLC) dataset to supplement the data (Karra et al., 2021). The EGLC dataset was produced on the basis of

**3.** In line 199, please clarify what "the rivers from 2020" refers to. If it refers to the extraction results from this study for 2020, please clarify the potential impact of generating validation samples based on extraction results on the randomness and representativeness of the samples.

**Resource 3:**

Thanks for your careful review and offer valuable comments. "the river from 2020" was the river map of CRED in 2020. We considered that CRED might omit some rivers, and generating random samples only within the CRED extent would make it difficult to evaluate these omission errors. Therefore, we spatially overlaid the 2020 CRED with the 2020 CNLUCC rivers. The union regions were used to create random river points. This procedure accounts for river identified in CNLUCC that are absent in CRED, allowing for more comprehensive assessment of omission errors.

For the generated random points, we conducted visual interpretation using imagery from different years between 2016 and 2023. These samples may have different attributes across different years, although their location remained unchanged. Thus, the generate samples were used not only to evaluate the accuracy of the 2020 CRED, but also for accuracy assessment of CRED in other years.

We have clarified the confused sentence, and provided more detailed description for generating validation samples.

> Specifically, the rivers in the CNLUCC dataset for 2020 were extracted and then overlaid with the rivers from 2020 of CRED.
>
> 220 In the union regions, river samples were created via random sampling. After that, non-river samples were produced within a
>
> 300 m outside buffer of the union regions. Second, all random samples were visually interpreted by combining Collect Earth
>
> (CE), Google Earth (GE) and the GEE platform (Peng et al., 2024b). The CE software enables user-friendly sample
>
> management, the GE provides high spatial resolution images, and the GEE offers median-composited Sentinel-2 images. We
>
> visually labelled these samples based on imagery from different year from 2016 to 2023. Correspondingly, their attributions
>
> 225 may vary across different years, although their locations remain unchanged. Using these three platforms, river and non-river
>
> samples from 2016 to 2023 were produced...

**4.** The sample size for validation is unclear. Please provide details on the distribution and quantity of the validation samples in Section 4.1.

**Response 4:**

Thanks for your useful comments. This suggestion is definitely helpful to enhance our manuscript. We have revised this point in our manuscript. The sample size from 2016 to 2023 and their spatial distribution was displayed in supplementary materials.

from 2016 to 2023. We validated the CRED via test samples that were manually inspected via visual interpretation. The sample sizes from 2016 to 2023 were shown in Table S1, and their spatial distribution was shown in Fig. S2. The CRED achieved

Table S1. Sample size of river and non-river from 2016 to 2023 in China

|  | 2016 | 2017 | 2018 | 2019 | 2020 | 2021 | 2022 | 2023 |
|---|---|---|---|---|---|---|---|---|
| River | 196 | 291 | 275 | 292 | 295 | 266 | 276 | 259 |
| Non-river | 652 | 597 | 585 | 572 | 542 | 586 | 567 | 596 |

[Figure]

Fig. S2. Spatial distribution of river and non-river samples from 2016 to 2023

**5.** The resolutions of the three existing products used for comparing river extraction results are not exactly the same. Did the authors perform any resampling or other processing when comparing river areas to eliminate the area differences caused by resolution?

**Response 5:**

Thanks for your careful review and offer professional comments. We did not perform any resampling or other processing when make data inter-comparison. In our study, three datasets—CNLUCC, CWaC and EA_Wetlands—were used to make data inter-comparison. The spatial resolution of CWaC and EA_wetlands is same as our CRED, with a spatial resolution of 10 m.

The CNLUCC, 30-m resolution datasets, had a lower spatial resolution compared to our CRED. Indeed, due to this spatial limitation, some narrow rivers that can be identified in CRED are not detected in CNLUCC. However, given the relatively similar spatial resolutions and the high achieved through manual visual interpretation, we included the CNLUCC in our comparative analysis. This procedure may contain some uncertainties into the comparative analysis. We described this point in the Discussion section.

> as by using the maximum, minimum or medium water inundated areas, is a debated issue. For the data inter-comparison
> between CNLUCC and CRED, there may be some uncertainties due to differences of spatial resolution. For examples, some
> 460   narrow streams are extracted in CRED but not in CNLUCC. Meanwhile, we did not use test samples to assess rivers' accuracy

**6.** In line 270, it is mentioned that the CRED dataset outperforms the existing most accurate products in extracting narrow rivers in mountainous areas. Did the authors consider providing a more precise definition of narrow rivers to highlight the advantages of this product?

**Response 6:**

Thanks for your careful review and offer specific suggestion. We added the definition of narrow rivers in the manuscript.

> their spatial disconnection. In our study, narrow rivers were defined as linear water bodies with a width greater than 10m.
> Generally, rivers with a width exceeding 30 m exhibited good spatial continuity in Sentinel-2 imagery, which could be
> automatically and accurately extracted by our algorithm. However, narrow rivers with width less than 30 m, due to spatial
> discontinuities, were challenging to be identified using geometric features. These narrow rivers were manually edited to
> 215   improve our rivers' accuracy.

**7.** The area difference mentioned in line 273 between the river areas of CWaC and the CRED in 2020 is inconsistent with the visualization results in Fig. 8. Please provide more detailed comparisons of the water bodies extracted result.

**Response 7**:

Thanks for your careful review and offering valuable comment.

In Figure 8, the river extents shown by CWaC appears larger than that of our CRED, mainly due to the display of vector data under different scale. For the CWaC, there are large amount of fragmented river waters, most of which are smaller than 1 ha. When displayed at the national scale, these fragmented waters are stack together. Even with a very small line width set for these fragmented river waters, their display at national scale is still evident. To illustrate this phenomenon more clearly, we mapped rivers with areas smaller and larger than 1 ha separately. It was found that CWaC has large amount of fragmented river waters, and its spatially continuous rivers are fewer than CRED.

[Figure]

Figure T4. The spatial distribution of two group of rivers in CWaC (2020)

To better visualize these small rivers, we present the spatial comparison of CWaC and CRED at the region scale in typical areas. The results indicate that in large-scale maps, fragmented rivers are stacked together. In small-scale maps, these waters are shown as individual small-area patches. Due to the above phenomenon, the rivers in CWaC are visually larger than those in CRED, but the river area counted in CWaC is smaller than in CRED. We have added these two figures to the supplementary material and described this phenomenon in the revised manuscript.

[Figure]

Figure T5. Spatial comparison of rivers between CRED and CEaC in 2020

For example, CWaC misclassified some reservoirs as rivers, whereas the CRED accurately excluded them (Figure S5 (F)). It should be noted that the rivers in CWaC appeared larger than these in CRED visually, mainly due to difference in display at different scale. To clearly illustrate this phenomenon, we mapped rivers of CWaC with areas less than and larger than 1 ha separately (Fig. S6).

---

## Author Comment (AC2)

**Response to The Comments from Reviewer #2**

This paper aims to produce a dataset of Chinese rivers spanning the period from 2016 to 2023 at an annual scale with a resolution of 10 m. However, the dataset lacks originality and has gaps in sufficient quality, and is limited in its potential for broader application, which I detail below:

**Response:**

Thank you very much for your precious time to review our manuscript. We sincerely appreciate your diligent efforts to provide professional comments. Referring to your comments, we have revised our manuscript and provided detailed explanations for each comment. We acknowledge that our paper has limitations, but we believe that it still is a meaningful and interesting study. Our river algorithm is accurate, robust and effective, which can achieve lower misclassification and omission errors compared to using only the length-to-with ratio. Meanwhile, considerable manual editions were implemented for our river maps, which further improve their data quality. The characteristics of being national-scale, annually continuous, and having a 10-m spatial resolution make our river maps valuable for practical applications. Our detailed responses are as follows.

**Originality:**

1) The classification of water body is more easily achievable compared to other land cover types in the field of remote sensing. It exhibits a significant spectral difference from other land cover types and has a relatively simple texture. Furthermore, it would be easy to screen rivers by simply using the length-to-width ratio of water bodies. However, the authors utilized publicly available 10 m land cover data and did not used an innovative scheme to extract rivers. They also fail to consider the network of rivers and the topographical features that influence the formation of rivers. The originality of the technical solution is limited.

**Response:**

Thanks for your thorough and professional comments.

As mentioned in the review comments, we did not develop a new algorithm for water classification. We did so mainly consider three aspects: (1) As the comment pointed out, the spectral features of water bodies are simple and differ significantly with other land use types. To date, there are a lot of studies on remote sensing classification of water bodies, and the related algorithms for classifying waters are mature, accurate and effective (Zou et al., 2018; Pickens et al., 2020; Yang et al., 2020). Developing a new algorithm would hardly lead to improvements in water classification, mainly due to the limitations of remote sensing image classification capabilities and the complexity of geographic environments. (2) The Dynamic World (DW) and ESRI global land cover (EGLC) used in this study are land use datasets generated using deep learning algorithm. The broad land use

types (e.g. water, forest, farmland) in these two datasets could achieve high accuracy using deep learning algorithm, especially for the waters. For DW tiles with considerable gaps in invalid observations in 2016, we used multiple index water detection rule (MIWDR) to extract water bodies. This method was proposed by Deng et al. (2019), which exhibited good feasibility for long-term water classification at large scale. Meanwhile, existing research has demonstrated that deep learning, machine learning and water index methods all can obtain reliable results (Li et al.,2022). Thus, we believe that all three datasets are reliable water-related products, with high accuracy of waters. (3) Considering the availability of mature water-relate datasets, which have been validated for accuracy and reviewed by experts, we directly used these datasets to produce water maps. This approach is not only to avoid duplicating work, but also to save a significant amount of workload.

**References:**

[1] Zou Z, Xiao X, Dong J, et al. Divergent trends of open-surface water body area in the contiguous United States from 1984 to 2016[J]. Proceedings of the National Academy of Sciences, 2018, 115(15): 3810-3815.

[2] Pickens, A. H., Hansen, M. C., Hancher, M., Stehman, S. V., Tyukavina, A., Potapov, P., Marroquin, B., & Sherani, Z. (2020). Mapping and sampling to characterize global inland water dynamics from 1999 to 2018 with full Landsat time-series. Remote Sensing of Environment, 243(March), 111792.

[3] Yang, X., Qin, Q., Yésou, H., Ledauphin, T., Koehl, M., Grussenmeyer, P., & Zhu, Z. (2020). Monthly estimation of the surface water extent in France at a 10-m resolution using Sentinel-2 data. Remote Sensing of Environment, 244(October 2019), 111803.

[4] Deng, Y., Jiang, W., Tang, Z., Ling, Z., & Wu, Z. (2019). Long-term changes of open-surface water bodies in the Yangtze River Basin based on the google earth engine cloud platform. Remote Sensing, 11(19).

[5] Li J, Ma R, Cao Z, et al. Satellite detection of surface water extent: A review of methodology[J]. Water, 2022, 14(7): 1148.

For our river extraction algorithm, it was accurate and effective for large-scale river mapping. On the one hand, it can accurately distinguish rivers from other water covers. On the other hand, the rules and thresholds used in our algorithm do not change over regions and time.

We believe our river mapping algorithm is an innovative algorithm, and our explanations is as follows: It is true that the length-to-width ratio can effective distinguish rivers from other water covers. However, using this single feature would result in considerable misclassification and omission errors. To illustrate this point, we counted length/width of rivers, lakes and reservoirs at different area groups. It was found that a threshold for the length/width set too low misclassifies

many lakes and reservoirs as rivers, whereas a threshold set too high leads to the omission of many rivers. To address this issue, we selected four geometric features—Compactness, Length/width, Roundness and Rectangular fit—based on literature review and experiments. To reduce omission errors, we first developed weak rules to extract as many as rivers as possible. Then, we combined these weak rules into strong rules to distinguish non-rivers such as lakes, reservoirs and aquaculture ponds, thereby reducing misclassification.

[Figure]

Figure T6. Violin plots and quartile box plots of length/width. The columns represent three area range groups of 0–1000 ha, 1000–5000 ha, and >5000 ha,

In addition to the computation of the river classification algorithm, considerable manual corrections were implemented to improve our river maps. There are three main errors need to be corrected: 1) Some aquaculture ponds were spatially connected with rivers. (2) The identified reservoirs may contain some rivers. (3) Small rivers with narrow channels cannot be well identified via the river rule set. We provided a more detailed description of post-processing improvement in Section 3.3. We also implemented considerable manual corrections. Therefore, our study is not a work that can be realized easily and has good potential in scientific research and practical applications.

2) As shown below, the authors did not acknowledge many relevant river datasets in the text. This makes me seriously concerned about the proper place for this paper.

**Response:**

Thanks for your careful review and offering professional comments.

As the comments mentioned, we indeed missed some typical and representative datasets in our literature review, which is a deficit of our paper. The three recommend datasets are wonderful global river datasets. We have added these datasets to our literature review in Section 5.3. Thanks again for your helpful comments. Additionally, we have re-searched the literatures in Web of Science platform, and added three additional global river datasets in our manuscript.

[1]   Lin, P., Pan, M., Wood, E. F., Yamazaki, D., & Allen, G. H. (2021). A new vector-based global river network dataset accounting for variable drainage density. Scientific data, 8(1), 28.

[2]   Nyberg, B., Sayre, R., & Luijendijk, E. (2024). Increasing seasonal variation in the extent of

rivers and lakes from 1984 to 2022. Hydrology and Earth System Sciences, 28(7), 1653-1663.

[3] Yan, D., Wang, K., Qin, T., Weng, B., Wang, H., Bi, W., et al. (2019). A data set of global river networks and corresponding water resources zones divisions. Scientific data, 6(1), 219.

[4] Yamazaki D, Ikeshima D, Sosa J, et al. MERIT Hydro: A high‑resolution global hydrography map based on latest topography dataset[J]. Water Resources Research, 2019, 55(6): 5053-5073.

[5] Yan D, Li C, Zhang X, et al. A data set of global river networks and corresponding water resources zones divisions v2[J]. Scientific Data, 2022, 9(1): 770.

[6] Altenau E H, Pavelsky T M, Durand M T, et al. The Surface Water and Ocean Topography (SWOT) Mission River Database (SWORD): A global river network for satellite data products[J]. Water Resources Research, 2021, 57(7): e2021WR030054.

■ **Table 4. List of existing river datasets covering the globe or China. The dashed lines indicate that the corresponding information is unavailable.**

420

| Type | Name | Spatial extent | Spatial resolution | Time period | Resource |
|---|---|---|---|---|---|
| River lines | GRRATS | Global | --- | --- | (Coss et al., 2020) |
| | HydroRIVERS | Global | 500 m | --- | (Lehner and Grill, 2013) |
| | MCRW | China | 30 m | 1990–2015 | (Yang et al., 2020a) |
| | Reach-level Bankfull river | Global | 30 m | --- | (Lin et al., 2020) |
| | Vector-based global river network | Global | 90 m | --- | (Lin et al., 2021) |
| | Global River Network | Global | 90 m | --- | (Yan et al., 2019) |
| | MERIT Hydro | Global | 90 m | --- | (Yamazaki et al., 2019) |
| | SWOT River Database | Global | 100 | --- | (Altenau et al., 2021) |
| River polygons | | | | | |
| | GLWD | Global | 1 km | 1980s | (Lehner and Doll, 2004) |
| | GRWL | Global | 30 m | --- | (Allen and Pavelsky, 2018) |
| | EA_Wetlands | East Asia | 10 m | 2021 | (Wang et al., 2023a) |
| | CAS_Wetlands | China | 30 m | 2015 | (Mao et al., 2020) |
| | CWaC | China | 10 m | 2020 | (Li and Niu, 2022) |
| | River_OSM | Global | --- | --- | https://download.geofabrik.de/asia/china.html |
| | Global River and Lake Extent | Global | 30 m | 1984-2022 | (Nyberg et al., 2024) |

3) Besides, the work of Allen & Pavelsky (2018) has been cited, but the differences from the data of this article have not been explained.

**Response:**

Thanks for your useful comments. To demonstrate the differences between GRWL and our CRED, we conducted additional experiments to perform a comparative analysis of the two datasets.

For GRWL and the CRED in 2018, the river areas were 25583.15 km$^2$ and 5186223 km$^2$, respectively. The river area in GRWL was significantly smaller than that in the CRED, which was also observed in Figure T7. The area difference of these two datasets was mainly due to the difference of river mapping methods. The GRWL only mapped rivers with width greater than 90 m, as it considered river extracted from Landsat imagery to be reliable only when their width exceeded 90 m. In contrast, the CRED mapped more narrow rivers, due to its high spatial resolution.

[Figure]

Figure T7. River patterns of GRWL and CRED in 2018

To quantify the consistency between GRWL and CRED, river fractions within a 0.05° by 0.05° grid were counted. It was found that the two datasets had moderate correlation coefficient, with R value of 0.577. This indicated that two datasets shared considerable consistency but still exhibited significant differences. Due to the significant difference in areas and spatial distribution between two datasets, we will no longer added the data inter-comparison between GRWL and CRED in the revised manuscript.

[Figure]

Fig. T8. Scatterplots of the river fraction between the CRED and GRWL

**Scientific quality:**

**1)** The scheme of data validation has considerable uncertainty. The river is typically characterized by the property of network morphology. However, the generated river data display a substantial number of river discontinuities. There is a conspicuous phenomenon of rivers in adjacent years either "disappearing" or "breaking off" noticeably. Although the accuracy is about 95% by visual interpretation, this is based on a pixel-by-pixel basis and does not take into account the connectivity of rivers. Additionally, the visual interpretation is also a highly subjective process. If only the center of the river was selected, the accuracy of the river would be overestimated.

**Response:**

Thanks for your valuable comments and careful inspection for our dataset.

The rivers are fluid linear waters with varied extents. Their surfaces area changes are affected by multiple factors, such as seasonal variation, climate changes and human activities. The CRED just mapped the actual extent of rivers in each year based on Sentinel-2 imagery. For the conspicuous phenomenon of rivers in adjacent years, most of them are attributed to natural factors and human activities. Meanwhile, there are still rivers inconsistencies between different years, which could be attributed to two aspects: (1) Some lakes, reservoirs and aquaculture ponds are spatially connected with rivers. For example, in the Poyang Lake Basin, lakes and rivers are spatially connected (Figure T9(a)). It is necessary to implement interrupt operation and delete lakes (Figure T9 (b)). However, since the interruptions are not strictly at the same location, this may lead to some inconsistencies in the rivers between years. (2) Many narrow rivers are spatially discontinuous, due to the limitation of 10 m spatial resolution. In our study, we made our best to manually these rivers. However, the corrections varied between different years, resulting in certain discrepancies in the CRED across years. To illustrate this limitation, we added corresponding descriptions in the Discussion section.

[Figure]

Figure T9. Spatial distribution of rivers without (a) and with (b) manual edition in 2023

smaller than that in other years. In addition, the extent and completeness of manual corrections varied across different years, leading to some inconsistencies of rivers in adjacent years. The above characteristics of the three datasets may lead to errors

In our accuracy validation, test samples were produced based on stratified random sampling and visual interpretation. On the one hand, the locations of test samples were not manually selected but were randomly generated. We did not deliberately produce river samples at the center of rivers. On the other hand, the attributions (river or non-river) were determined based on visual interpretation. This approach ensures both a reasonable sample distribution and accurate sample attributions. We have made more detailed descriptions for the production of test samples.

To assess the mapping accuracy of rivers, test samples were produced by combining stratified random sampling and visual interpretation. First, river and non-river samples were produced via overlay, buffer and random sampling operations. Specifically, the rivers in the CNLUCC dataset for 2020 were extracted and then overlaid with the rivers from 2020 of CRED. 220 In the union regions, river samples were created via random sampling. After that, non-river samples were produced within a 300 m outside buffer of the union regions. Second, all random samples were visually interpreted by combining Collect Earth (CE), Google Earth (GE) and the GEE platform (Peng et al., 2024b). The CE software enables user-friendly sample management, the GE provides high spatial resolution images, and the GEE offers median-composited Sentinel-2 images. We visually labelled these samples based on imagery from different year from 2016 to 2023. Correspondingly, their attributions 225 may vary across different years, although their locations remain unchanged. Using these three platforms, river and non-river samples from 2016 to 2023 were produced.

**2)** The key data utilized in this paper (i.e., European Space Agency and Dynamic World) are 10 m land-cover classification products. They were not primarily designed for water classification. These products tend to underestimate the area of water body, and consequently, the extent of rivers.

**Response:**

Thanks for your valuable comments.

To date, there are many global or national water datasets. However, the long-term and 10 m spatial resolution at national or global scale still remain scare. The DW dataset is a real-time updated land use dataset since 2015. Its high temporal resolution allows it to effectively capture river dynamics in rivers, while its 10 m spatial resolution enables accurate extraction of rivers, as well as narrow rivers.

The DW datasets were produced using deep learning algorithms, and their reliability has been validated by accuracy assessment and peer-reviewed. Additionally, Li et al (2022) indicated that deep learning algorithms, machine learning algorithms and water index threshold algorithms all achieve good accuracy in water classification. Although the DW were not primarily designed for water classification, rivers in the DW also achieved high accuracy, making them reliable for river

mappings. It is true that the use of DW land use data for river extraction may cause uncertainties. We have illustrated this point in the Discussion section.

> In future studies, the data quality and time range of the CRED could be further improved. As previously discussed, many
> 465     uncertainties in the CRED are largely caused by multisource LUCC datasets or water datasets. Despite the availability of water
> datasets at 10 m spatial resolution(Li et al., 2023; Vanderhoof et al., 2023; Yang et al., 2020b), long-term global water datasets
> are still scarce. Thus, future attempts should focus on developing robust and accurate water classification algorithms for

Reference:

[1]     Li J, Ma R, Cao Z, et al. Satellite detection of surface water extent: A review of methodology[J]. Water, 2022, 14(7): 1148.

**3)** Rivers possess highly pronounced seasonal characteristics. During the summer flood season, rivers become wider, while in winter, they may even disappear. The specific meaning and significance of annual-scale river data remains unclear.

**Response:**

Thanks for your professional comments.

How to define rivers is a challenging task. During the wet season, the extents of river inundation is large, part of which may consist of floodwaters. During the dry season, the extents of river is small, which underestimate the actual extent of rivers. To balance this contradiction, we map river extents using the mode algorithm based on water time series with one year. For a pixel of water, if most of values in its time series are water, the pixel is labeled as yearly water. In our study, the yearly water of rivers is defined as river extents. To explicitly descript the definition of rivers, we made supplements in our manuscript.

> value of this pixel is 0. In this way, annual water maps were created via the mode algorithm. For rivers, the composited water
> extents were defined as yearly river extents. Additionally, considering the scarcity of DW images before 2016, yearly water

**4)** "For areas with missing DW data, the EGLC and Sentinel-2 images were chosen as supplementary datasets, which were utilized to create annual water maps." This strategy is subjective. As depicted in Figure 2, the EGLC data only encompasses the period from 2017 to 2023. In contrast, for the remaining years of 2015 - 2016, classification is carried out using the land cover data that was self-produced. Why use the DW dataset as the primary data of river extraction? How to ensure consistency across datasets? The experimental scheme also has a certain degree of subjectivity.

**Response:**

Thanks for your constructive comments. This is very professional comments.

The Dynamic World (DW) is a 10m near-real-time land use datasets. Normally, the revision of

DW is 3-5 days. This high-frequency is beneficial for capturing river dynamics. Thus, we used the DW as primary data to extract rivers. However, the DW dataset was produced using Sentinel-2 images with cloud coverage less than 35%. This strategy leads to sparse or even absent valid observations in areas severely affected by cloud contaminations.

The following figure counts the number of good observations of 2023 DW in each pixel. Results indicated that there are fewer or missing good observation pixels in Southwest China. To solve this issue, we used EGLC dataset to replace DW dataset in regions with spare or missing observations. However, the time range of EGLC is 2017-2023. To completely produce river maps in 2016, the Sentinel-2 images were used to extract waters in areas with spare or invalid observations of DW. Considering the limited availability of Sentinel-2 images before 2017, all Sentinel-2 images from June 2015 to December 2016 were utilized for water classification.

[Figure]

Figure T10. Valid observations for individual pixels in DW image of 2023. This figure was also displayed in Supplementary material

To explicitly illustrate the strategy for the selection of three datasets, we have added supplementary explanations in the manuscript. In addition, we agree that using three different data sources to produce river maps would introduce uncertainties. For instance, river extents may vary in same location and time when derived from different datasets. This is a limitation of our study, which we have described in the Discussion section.

90  areas had high mapping accuracy. Statistical analysis revealed that many regions in China had no or few good observations (Fig. S1). This occurred because the DW images were produced from Sentinel-2 images with less than 35% cloud coverage. Sentinel-2 images used in DW dataset may have could contaminations, which also resulted in missing observations. Thus, the complete coverage of the entire China to produce a complete China river map cannot be achieved by using DW dataset alone.

    To facilitate image computation, we divided China into 52 tiles using a 5°×5° grid (Fig. S1). If a tile contains considerable

95  pixels with fewer than 3 valid observations, it was defined as a data insufficient tile. For these tiles, we used the ESRI global land cover (EGLC) dataset to supplement the data (Karra et al., 2021). The EGLC dataset was produced on the basis of Sentinel-2 images via deep learning algorithms. It recorded annual global land use types from 2017 to 2023 with high accuracy for water bodies. In addition, we also collected Sentinel-2 images to produce water maps in areas missing data from the DW dataset in 2016 because of the lack of 10-m LUCC or water datasets. All Sentinel-2 images from 2015 to 2016 were used for

100  water classification.

465  In future studies, the data quality and time range of the CRED could be further improved. As previously discussed, many uncertainties in the CRED are largely caused by multisource LUCC datasets or water datasets. Despite the availability of water datasets at 10 m spatial resolution(Li et al., 2023; Vanderhoof et al., 2023; Yang et al., 2020b), long-term global water datasets are still scarce. Thus, future attempts should focus on developing robust and accurate water classification algorithms for

**Application**

**1)** As illustrated in Fig. 8, the river data produced in this paper is significantly different from that of other products. In practical applications, it is relatively difficult for users to make a trade-off regarding which one to use.

**Response:**

  Thanks for your careful and offering professional comments.

  In our study, we check the quality of river datasets based on spatial comparison by visual inspection. It was found that, for EA_Wetlands, it mapped more rivers compared with CRED. However, it contained considerable misclassifications. For example, the Poyang Lake that spatially connected with Yangtze River was labeled as rivers. Aquaculture ponds that connected with rivers were labeled as rivers. For CWaC, many small patches, such as snow/ice, mountain shadows, and other small waters, were misclassified as rivers. For CNLUCC, it failed to identify narrow rivers due to the 30m spatial resolution limitation. In contrast, our CRED not only accurately rivers with fewer misclassifications, but also effectively identified narrow rivers. Based on above analysis, we conclude that CRED not only accurately mapped river extents in China, but also reflect temporal changes of rivers. Therefore, CRED can serve as reliable dataset for river-related applications.

  We implemented data inter-comparison for two purposes. On the one hand, it is to show that our data have good agreement with existing river-relate datasets. On the other hand, spatial inspections were implemented to analysis the strengths and weakness of different datasets, which

could highlight the superiority of the CRED.

In our study, we did not implement strict accuracy validation for different river datasets, and cannot quantitatively assess the quality of river datasets. This is an insufficient of our work. We illustrated this point in the Discussion section.

> narrow streams are extracted in CRED but not in CNLUCC. Meanwhile, we did not use test samples to assess rivers' accuracy of CNLUCC, CWaC and EA_Wetlands, which cannot quantitatively evaluate the accuracy superiority of CRED compared to other river datasets. In addition, our CRED maps were national-scale dataset, and its applicability was not as extensive as

2) This is not a global product. It has relatively limited application potential compared with other global river products.

**Response:**

Thanks for your valuable comments.

It is true that our CRED is not global river dataset, which to some extent limits its widespread application. However, compared with other large-scale river datasets, our CRED has its own advantages. First, our algorithm is accurate and efficient for river extracted, and the identified rivers have small errors in omission and misclassifications. Second, extensive post-processing was conducted on the algorithm-extracted rivers. Aquaculture ponds connected with rivers were manually interrupted and then removed. Lakes and reservoirs connected with rivers were also manually corrected. Narrow rivers with spatial discontinuous patches were manually corrected to map rivers as much as possible. Thus, river boundaries in the CRED are accurate and complete, and the CRED data has good application potentials in China.

Considering the huge workload and our limited time, we did not produce global river dataset. We acknowledge that the river datasets at the national scale has limited applicability compared to global-scale data. This point is illustrated in the discussion section.

> other river datasets. In addition, our CRED maps were national-scale dataset, and its applicability was not as extensive as global-scale dataset.